# Pre-saccadic remapping relies on dynamics of spatial attention

Martin Szinte[1]*, Donatas Jonikaitis[2], Dragan Rangelov[3], Heiner Deubel[4]

[1]Department of Cognitive Psychology, Vrije Universiteit, Amsterdam, The Netherlands; [2]Department of Neurobiology, Howard Hughes Medical Institute, Stanford University School of Medicine, Stanford, United States; [3]Queensland Brain Institute, The University of Queensland, Brisbane, Australia; [4]Allgemeine und Experimentelle Psychologie, Ludwig-Maximilians-Universität München, Munich, Germany

**Abstract** Each saccade shifts the projections of the visual scene on the retina. It has been proposed that the receptive fields of neurons in oculomotor areas are predictively remapped to account for these shifts. While remapping of the whole visual scene seems prohibitively complex, selection by attention may limit these processes to a subset of attended locations. Because attentional selection consumes time, remapping of attended locations should evolve in time, too. In our study, we cued a spatial location by presenting an attention-capturing cue at different times before a saccade and constructed maps of attentional allocation across the visual field. We observed no remapping of attention when the cue appeared shortly before saccade. In contrast, when the cue appeared sufficiently early before saccade, attentional resources were reallocated precisely to the remapped location. Our results show that pre-saccadic remapping takes time to develop suggesting that it relies on the spatial and temporal dynamics of spatial attention.
DOI: https://doi.org/10.7554/eLife.37598.001

*For correspondence:
martin.szinte@gmail.com

**Competing interests:** The authors declare that no competing interests exist.

## Introduction

Our eye movements shift the visual scene on our retinas. These shifts go largely unnoticed and do not prevent efficient interaction with objects surrounding us. It has been proposed that the visual system compensates for such shifts using a copy of the motor command (*Sperry, 1950*) to anticipate changes in the visual scene from the planned eye movement. Such an active mechanism could maintain an impression of space constancy and allow us to effectively interact with visual objects. However, we typically do not keep track of the whole visual scene (*O'Regan et al., 1999*; *Rensink et al., 1997*). Studies have proposed that such visual compensation could be restricted to salient or task-relevant objects, selected by spatial attention (*Cavanagh et al., 2010*; *Rolfs and Szinte, 2016*). At the behavioral level, this compensation could result in anticipatory deployment of spatial attention to the retinal location that a visual stimulus will occupy after the saccade (*Jonikaitis and Theeuwes, 2013*; *Rolfs et al., 2011*; *Szinte et al., 2015*; *Szinte et al., 2016*). Such anticipatory deployment could explain observations that attention is allocated at a spatial target location almost immediately after a saccade (*Jonikaitis et al., 2013*; *Yao et al., 2016b*).

At the neuronal level, these visual compensations have been described as a remapping of visual neuron receptive fields. Remapping triggers an anticipatory and, sometimes, pre-saccadic response of neurons in frontal eye fields (FEF), lateral intra-parietal area (LIP) and superior colliculus (SC) with receptive fields centered on the post-saccadic retinal location of the attended object (*Duhamel et al., 1992*; *Sommer and Wurtz, 2006*; *Walker et al., 1995*). Remapping can facilitate tracking of task-relevant objects across saccades and allow rapid comparison between pre- and post-saccadic visual inputs (*Crapse and Sommer, 2012*). However, this remapping hypothesis has

been challenged with new data collected within the FEF (*Chen et al., 2018*; *Zirnsak and Moore, 2014*). These studies found that, before a saccade, neurons respond to stimuli presented near the saccade target rather than to stimuli presented at remapped locations of the recorded receptive field (RF). These results were later termed 'convergent remapping' towards the saccade target in dissociation of the 'forward remapping', which would be parallel to the saccade vector (*Neupane et al., 2016a*). They led to the proposal that convergent remapping could manifest behaviorally as a spatially unspecific spread of attention around the saccade target (*Zirnsak and Moore, 2014*). Remapping of spatial attention before saccades, as reported in behavioral studies, therefore could be reinterpreted as attentional spread between saccade target and remapped location (*Jonikaitis et al., 2013*; *Rolfs et al., 2011*; *Szinte et al., 2015*; *Szinte et al., 2016*). Such interpretation of the convergent remapping effects predicts that locations surrounding the saccade target by up to 10 degrees of visual angle (dva) would receive all attentional benefits before the eyes start to move. To date, there are no behavioral studies mapping pre-saccadic attention in sufficient detail to disambiguate whether attention converges towards the saccade target, or is remapped in parallel to the saccade target, as earlier behavioral work suggested.

We developed a protocol that allowed us to measure detailed maps of pre-saccadic attention, by measuring the orientation sensitivity at multiple locations while participants prepared a saccade (*Figure 1*). We observed that attention was allocated to the saccade target location and did not spread to the nearby positions, about 4.2 degrees of visual angle (dva apart. Next, we measured remapping of attention in the presence of a salient cue during a saccade task, manipulating the timing of the cue relative to the saccade. Our reasoning was that if remapping is an attentional process (*Cavanagh et al., 2010*; *Rolfs and Szinte, 2016*), it will take some time for the attention shift to

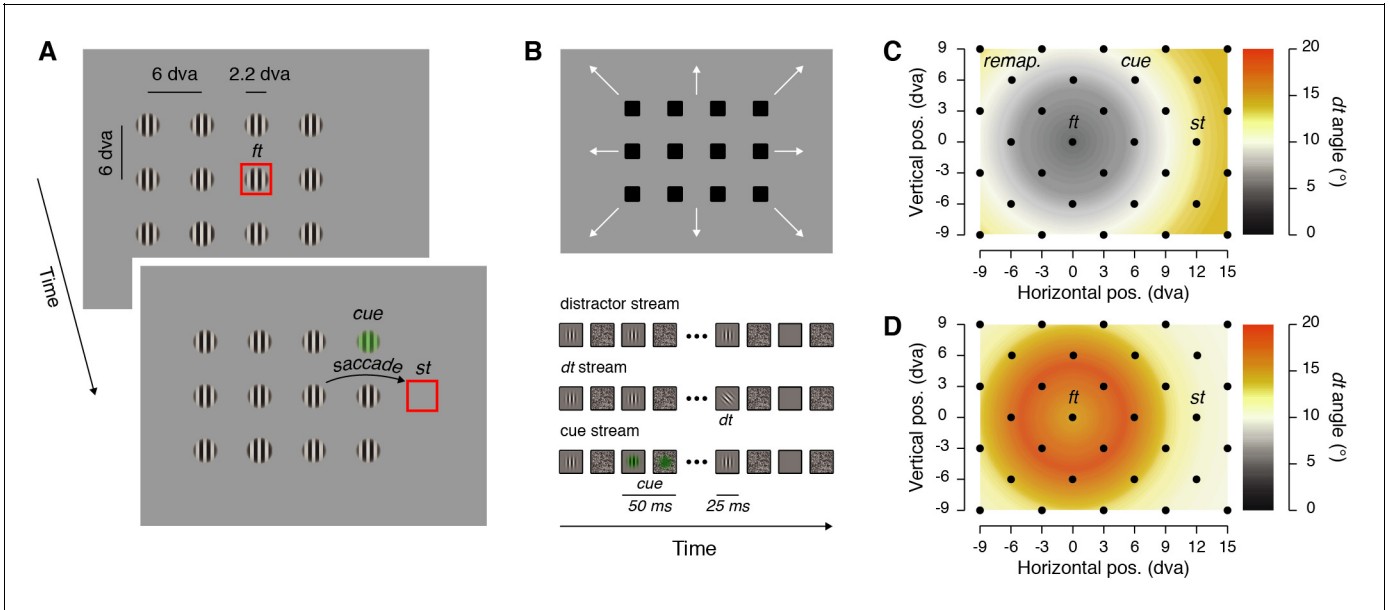

**Figure 1.** Stimulus displays and stimulus eccentricity effects. (**A**) Participants fixated on the fixation target (*ft*) and prepared a saccade towards the saccade target (*st*) presented either to the right or to the left of the fixation between 700 and 900 ms after the trial onset. On each trial, 12 visual streams (40 Hz flickering vertical Gabors and masks) were shown and in two out of the three trials a cue was flashed (50 ms) either above or below a virtual line between the fixation and the saccade targets (note that the stimuli here are sketched to increase their visibility, actual stimuli match those shown in the visual stream depiction). (**B**) The arrangement of visual streams can take several positions (see Materials and methods), to cover the whole display across trials. Participants reported the orientation of a discrimination target (*dt*), a tilted Gabor, presented within all trials at a time maximizing the occurrence of its offset within the 150 ms preceding the saccade. (**C–D**) The discrimination target was shown across trials at 32 different positions (see black dots) covering 24 dva horizontally and 18 dva vertically and including four main positions of interest (the fixation target: *ft*; the saccade target: *st*, the cue: *cue*; and the remapped location of the cue: *remap.*). The tilt of the discrimination target was titrated to yield comparable performance at differently cued eccentricities from the fixation target. We adjusted these tilts in a preliminary task made either while participants kept their eyes steady at the fixation target (**C**) peripheral remapping threshold task (see *Materials and methods*), or prepared a saccade (**D**) foveal remapping threshold task (see *Materials and methods*). The maps show *dt* tilt angles averaged across participants in these two threshold tasks.
DOI: https://doi.org/10.7554/eLife.37598.002

occur (*Ling and Carrasco, 2006*; *Müller and Rabbitt, 1989*; *Nakayama and Mackeben, 1989*; *Rolfs and Carrasco, 2012*). Therefore, stimuli presented just before a saccade would not leave enough time for remapping to develop and to be observed before the saccade. On the other hand, stimuli presented early enough should be remapped before the saccade. Indeed, we found that when the cue appeared shortly before saccade onset, spatial attention was allocated at the cued location but not at its remapped location. In contrast, when the cue appeared sufficiently early before saccade onset, attentional resources that were initially drawn to the cued location were re-allocated to its remapped location (i.e. the retinal location it will occupy after the saccade).

## Results

We determined spatially detailed maps of attention before a saccade under two different conditions: first, when participants made a visually guided saccade, and second, when a transient peripheral stimulus, a cue, was additionally presented during its preparation. We assessed spatial attention by asking participants to report the orientation of a briefly presented tilted discrimination target (clockwise or counterclockwise tilted Gabor), embedded in a display of vertical distractor streams (vertical Gabors, *Figure 1A–B*). To ensure that the discrimination task could be solved correctly only if participants attended at a particular location, we first completed a threshold task in which participants fixated at the center of the screen. This threshold task was used to estimate the tilt angle of a cued discrimination target presented at different eccentricities from the fixation. We observed that to achieve comparable discrimination at different eccentricities, the discrimination target had to be tilted by 4.42 ± 0.86 ° (mean ± SEM), if presented at the fixation target. This tilt gradually increased with eccentricity, finally reaching 14.10 ± 1.40 ° at eccentricities between ~15.3 and~16.2 dva (see *Figure 1C*). We used these threshold tilt values at their respective eccentricities in the main saccade task.

We first verified that presentation of the discrimination target during saccade preparation did not disrupt eye movements. Such a disruption, as measured by saccade latency or accuracy, would suggest that the stimuli used to measure attention instead captured attention. For this we first determined whether eccentricity of the discrimination target affected saccade latency. Saccade latency was longer when the visual streams overlapped with the fixation and saccade targets (217.56 ± 3.77 ms) compared with when they didn't overlap (186.00 ± 2.61 ms, $p < 0.0001$). This indicates that such difference resulted from the saccade target and fixation being less visible if they overlapped with the visual streams. Therefore, we separated the trials based on whether the fixation and saccade targets overlapped with the visual streams or not. Discrimination target eccentricity did not affect saccade latency on trials in which the fixation and saccade targets overlapped with visual streams. We did not observe a main effect of the discrimination target eccentricity (see Materials and methods for the definition of eccentricity), either for trials in which the fixation and saccade target overlapped with the visual streams (repeated measures ANOVA for four eccentricity groups used, $F_{3,39} = 0.08$, $p = 0.9725$, $\eta_p^2 3,39 = 0.03$ %) or for trials in which they didn't (for four eccentricity groups used, $F_{3,39} = 1.49$, $p = 0.2312$, $\eta_p^2 3,39 = 0.83$ %). Note that from a pilot study, we expected to find such saccade latency costs when the targets overlapped with the visual streams. To compensate for these expected effects, on trials in which the visual streams overlapped with the fixation and the saccade target, we presented the discrimination target 25 ms later than when there was no overlap. This procedure ensured homogenous timing of the discrimination target relative to the saccade onset irrespective of the tested position. This also ensured that any spatially unspecific increase of the discrimination threshold before a saccade (Campbell & Wurtz, 1978) could not explain differences between our experimental conditions and that discrimination performance truly reflected sensitivity gathered at the same instant relative to the saccade onset. Next, we evaluated whether discrimination target eccentricity (i.e. the absolute distance between the saccade target and the saccade landing point) affected saccade accuracy, as would be evident if the target captured attention. Again, we didn't find a main effect of the discrimination target eccentricity on the saccade accuracy (between five eccentricity groups used, $F_{4,52} = 2.11$, $p = 0.0929$, $\eta_p^2 3,39 = 1.38$ %). Altogether, these results show that presentation of the discrimination target did not disrupt saccade preparation, demonstrating that the stimuli we used to measure the allocation of attention did not directly interfere with its deployment.

Next, we measured the pre-saccadic allocation of attention. In all conditions, we analyzed performance to the presentation of discrimination targets within the 150 ms preceding the saccade. This procedure ensured that discrimination performance reflected the modulation of spatial attention over space rather than visual acuity and ensured that the same discrimination targets were matched across all conditions and trials. Trials with and without a cue were analyzed separately. On trials without a cue (*Figure 2A–C*), we found increased visual sensitivity (0.88 ± 0.05, normalized d' and SEM, respectively) at the saccade target location relative to the average of all other tested positions (0.42 ± 0.04, $p < 0.0001$, *Figure 2B*), suggesting that attention shifted towards the saccade targets during saccade preparation. Further, we tested the spatial specificity of this effect by comparing visual sensitivity at the saccade target location with the average visual sensitivity at the four positions surrounding it. Attention at the saccade target clearly did not spread to the surrounding positions (0.40 ± 0.04, $p < 0.0001$, *Figure 2C*), with sensitivity benefits being constrained to the immediate vicinity of the saccade target. Also, we did not observe a deployment of spatial attention to the fixation target (0.41 ± 0.05) compared with the average across all other positions ($p = 0.8952$).

Next, we analyzed the trials during which we presented an additional cue during saccade preparation, with the cue and discrimination target shown shortly after each other (50 ms) and on average 96.88 ± 0.96 ms (cue offset relative to saccade onset) before the saccade (*Figure 2D–F*). Here, in

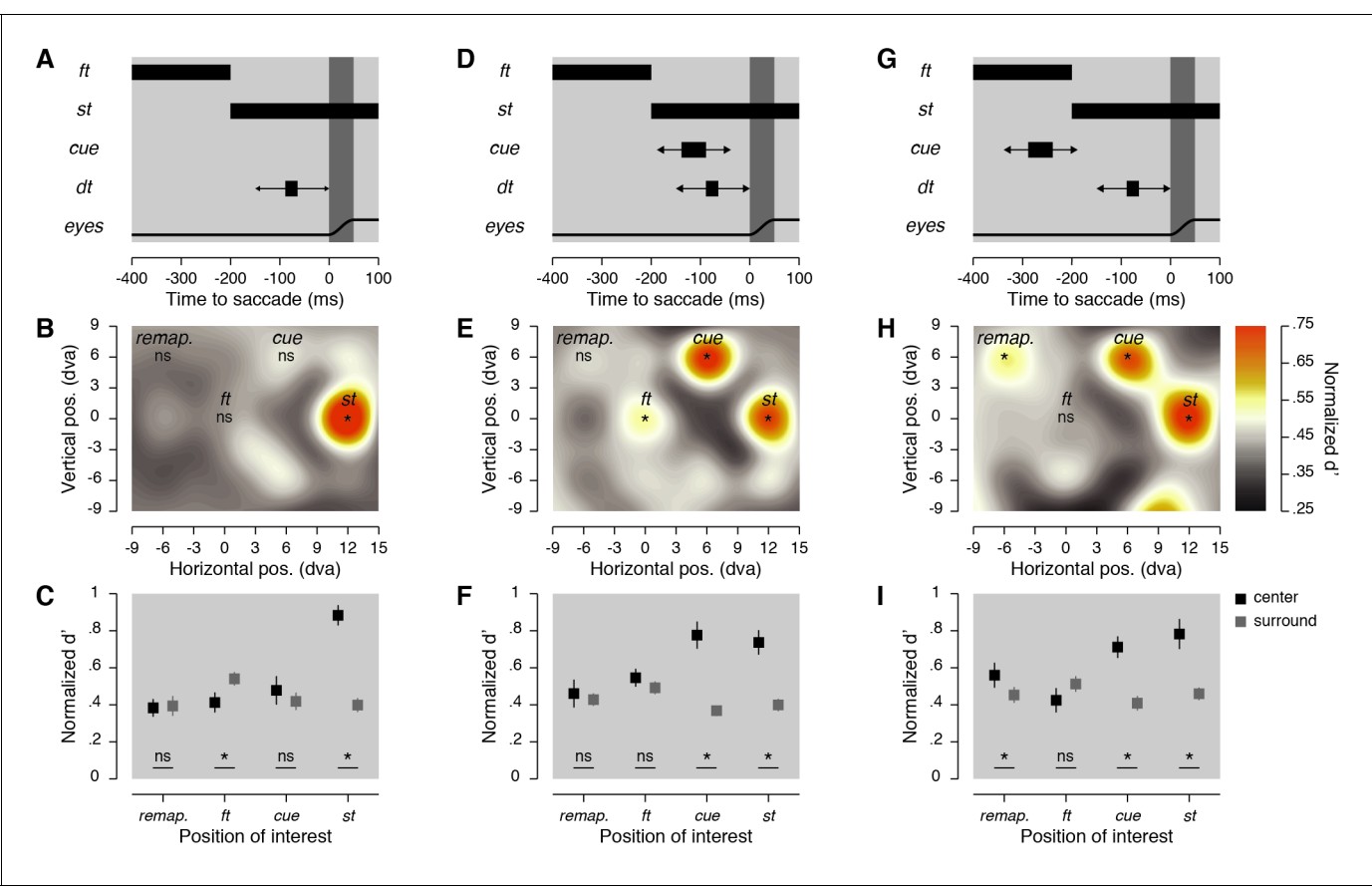

**Figure 2.** Stimulus timing and sensitivity maps. (A,D,G) Stimulus timing. Participants prepared a saccade at the offset of the fixation target (*ft*), which corresponded to the onset of the saccade target (*st*). In the 150 ms before the saccade, a discrimination target (*dt*) was briefly shown at one of the 32 possible positions. Then, no cue was shown (A), or a cue was shown 50 ms before the *dt* and about 100 ms before the saccade (D), or a cue was shown 200 ms before the *dt* and about 250 ms before the saccade (G). (B,E,H) Normalized sensitivity maps. Averaged normalized sensitivity (d') observed across participants and displayed using a color-coded linear scale going between 0.25 and 0.75 (see Materials and methods). Asterisks indicate significant differences ($p < 0.05$) in sensitivity found between a particular position of the *dt* and the average of all the other tested positions. (C,F,I) Averaged normalized d' obtained at four positions of interest (black squares) and at their corresponding surrounding positions (dark gray squares). Error bars show SEM and asterisks indicate significant comparisons ($p < 0.05$).

DOI: https://doi.org/10.7554/eLife.37598.003

addition to the fixation and saccade targets, we were also interested in two further locations: the cue location and the retinal location the cue will occupy after the saccade, that is, the remapped location. Visual sensitivity was higher at the cue (0.78 ± 0.07, *p* = 0.0002), at the saccade target (0.74 ± 0.07, *p* < 0.0001) and at the fixation target (0.55 ± 0.05, *p* = 0.0060), when compared to the average of all the tested locations (0.44 ± 0.03). When compared to their closest surrounding positions (*Figure 2F*), we found spatially specific effects only at the cue (surround: 0.37 ± 0.03, *p* < 0.0001) and at the saccade target (surround: 0.40 ± 0.03, *p* < 0.0001), but not at the fixation target (surround: 0.49 ± 0.03, *p* = 0.1112). Importantly, when the cue was shown shortly before the saccade, the visual sensitivity at its remapped location (0.46 ± 0.07) was not significantly higher relative to the other tested positions (0.44 ± 0.03, *p* = 0.6886). Thus, we observed no evidence for attentional remapping of the cued location when the cue appeared shortly before saccade onset.

These results contrasted with those found in trials when the same cue was shown substantially before the discrimination target (200 ms) and on average 240.82 ± 1.42 ms before the saccade onset (*Figure 2G–I*). Under such conditions, when compared with the average across all positions (0.45 ± 0.08), we found higher visual sensitivity at the saccade target (0.78 ± 0.08, *p* = 0.0004), at the cue (0.71 ± 0.06, *p* < 0.0001) and, critically, at the remapped location of the cue (0.56 ± 0.07, *p* = 0.0072). Similar to other experimental conditions, the benefits observed at the saccade target (surround: 0.46 ± 0.03, *p* = 0.0010), at the cue (surround: 0.41 ± 0.04, *p* < 0.0001) and at its remapped location (surround: 0.45 ± 0.04, *p* = 0.0382) did not spread towards their respective adjacent positions.

Finally, we found that the increase in sensitivity observed at the remapped location of the cue was present for the condition in which the cue appeared substantially before the discrimination target and the saccade onset, but not if it appeared later. Such an effect was evident from comparison of normalized sensitivity obtained at the remapped position of the cue in the condition in which no cue was shown (0.38 ± 0.05) to conditions in which the cue appeared substantially before the discrimination target and the saccade onset (0.56 ± 0.07, *p* < 0.0038) or to conditions in which it appeared later (0.46 ± 0.07, *p* = 0.2816). These comparisons can be visualized by mapping subtraction of the normalized sensitivity obtained in the conditions in which we displayed a cue from those in which no cue was shown (see *Figure 3* and Materials and methods). We normalized these differences, to present data with the same color scale as in the condition maps (*Figure 2B–2E–2H*). Combined, our results so far show that when the visual system is given enough time to process and attend a visual stimulus, such as the salient cue used in our task, spatial attention is remapped to the retinal location the stimulus will occupy after a saccade.

If attention is remapped pre-saccadically, one should also expect to find spatially specific attentional effects at the fixation target, as the fixation target is the remapped location of the saccade target. Indeed, one study reported such foveal remapping of the saccade target (*Rolfs et al., 2011*). In the experiment above, we observed inconsistent evidence for spatial attention at the fixation target (see *Figure 2C, F and I*). This is likely to be because of the threshold procedure differences between

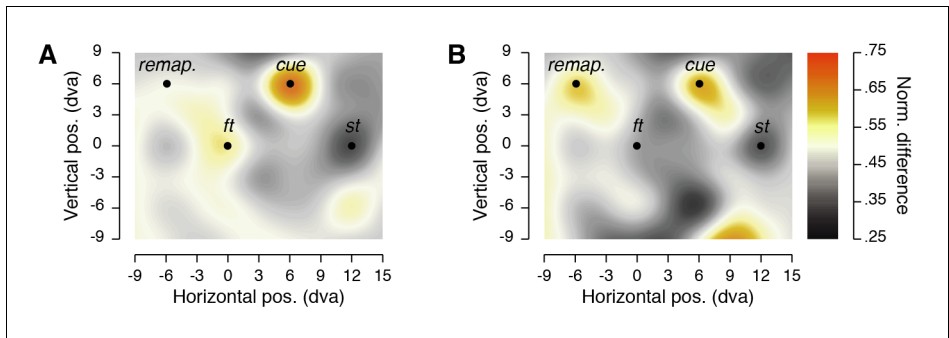

**Figure 3.** Cue vs. no-cue subtraction maps. Individual normalized sensitivity (d') is subtracted between conditions and the difference is normalized to obtain maps with convention as in *Figure 2*. Subtractions are made between trials in which the cue was shown ~100 ms before the saccade (**A**) or ~250 ms before the saccade (**B**) to trials in which it was not shown.
DOI: https://doi.org/10.7554/eLife.37598.004

this study and the one that reported spatial remapping of attention to fixation (*Rolfs et al., 2011*). Indeed, as we were principally interested in remapping of the salient peripheral cue in the above study, our threshold procedure measured spatial attention during fixation (yielding, importantly, no threshold difference between the cue and the remapped location of the cue). However, previous research has shown that preparation of saccades draws spatial attention away from other, non-saccade or non-salient locations (*Jonikaitis and Theeuwes, 2013*), resulting in an increase of acuity threshold in between the fixation and the saccade target locations (*Campbell and Wurtz, 1978*). In the experiment described above, we measured perceptual thresholds in a fixation task, rather than in a saccade task, so it is possible that the threshold for the fixation location underestimated that during saccadic preparation. If this was the case, potential remapping of the saccade location to the fixation location could have been masked by drawing attention away from the fixation location in the threshold task. Therefore, in the threshold task of a second experiment (foveal remapping task), we adjusted the tilt of the discrimination target using a saccade rather than a fixation task (see foveal remapping main task in *Materials and methods*). We observed that the discrimination target had to be tilted by 15.03 ± 2.34° if presented at the fixation target before a saccade (*Figure 1D*), a tilt up to four times bigger than that recorded during the fixation threshold procedure (*Figure 1C*). We used these values in a simplified version of the above experiment, without presentation of the cue, and as before, with a discrimination target randomly shown across trials at 32 possible positions while participants prepared a visually guided saccade. If anything, this procedure should be even more sensitive at detecting attention spread between saccade and fixation targets (*Zirnsak and Moore, 2014*).

Then in this second task, in which only the thresholding procedure was changed, we first analyzed whether the discrimination target affected saccade latency and accuracy. We found a main effect of the eccentricity of the discrimination target on saccade latency within trials in which the visual streams overlapped with the fixation and the saccade targets ($F_{3,21}$ = 8.89, $p$ = 0.0005, $\eta_p^2$3,21= 4.84 %), but not on trials when the visual streams did not overlap with the targets ($F_{3,21}$ = 3.01, $p$ = 0.0532, $\eta_p^2$3,21= 2.03 %). Further, discrimination target eccentricity did not affect saccade accuracy ($F_{4,28}$ = 1.60, $p$ = 0.2007, $\eta_p^2$3,21= 2.53 %). These results suggest that, similar to the first experiment, presentation of the discrimination target had limited influence on preparation of the saccade. Next, and in agreement with the results of the first experiment, we found a systematic pre-saccadic deployment of attention towards the saccade target (Figure 4A, 0.68 ± 0.11, $p$ < 0.0001) when compared with the average over all the tested positions (0.36 ± 0.03). Critically, this benefit was accompanied by systematic deployment of attention at the fixation target (0.96 ± 0.02, $p$ < 0.0001). Finally, these effects were spatially specific (Figure 4B), as shown by the significant differences observed when comparing sensitivity at the fixation target (surround: 0.47 ± 0.06, $p$ < 0.0001) and the saccade target (surround: 0.33 ± 0.04, $p$ < 0.0001) with their relative surrounds.

## Discussion

We constructed spatial attention maps by measuring orientation sensitivity while participants prepared a visually guided saccade. Our paradigm allowed us to measure whether attention spreads broadly around the saccade target or shifts towards spatially specific loci. We observed that attention consistently shifted to the saccade target location and, importantly, did not spread to other locations surrounding it. In our main manipulation, we presented a salient cue during saccade preparation. On these trials, we observed a second spatially specific locus of attention at the cued location. Importantly, on these cued trials, we also observed a third, distinct locus of attention. Although this third location was neither salient, nor task-relevant before the saccade, it corresponded to the retinotopic location the cue would occupy after the saccade. In other words, we observed attentional remapping of the cue location before the saccade onset. Critically, these effects were present only when the cue appeared long enough before the saccade onset. This indicates that remapping, like any other attentional process, requires some time. As we observed three separate foci of attention (at the saccade target, cue and remapped location), we did not find support for the hypothesis that attention spreads during saccade preparation around the saccade target (*Zirnsak and Moore, 2014*).

Our findings speak to the current debate on whether forward remapping exists and what role it plays in maintaining visual stability. As we report behavioral results, it is important to note that the link between our effects and neurophysiology is only theoretical. Behavioral experiments do not allow for direct conclusions as to which neural structures are involved in remapping nor on the validity of neurophysiology findings. However, our findings are relevant to the behavioural predictions provided in earlier neurophysiology work. Combined behavioral and neuronal recordings are necessary to eventually link proposals of neuronal activity before saccades and behavioral observations reported here and previously.

Behavioral studies on visual remapping of attention (*Jonikaitis et al., 2013*; *Rolfs et al., 2011*; *Szinte et al., 2015*; *Szinte et al., 2016*; *Yao et al., 2016b*) have been inspired by earlier neurophysiology work suggesting that receptive fields in FEF, LIP and SC shift (or are remapped) in anticipation of post-saccadic stimulus location (*Duhamel et al., 1992*; *Sommer and Wurtz, 2006*; *Walker et al., 1995*). Such a predictive receptive field shift occurs either shortly after a saccade (at a neural latency too short for visual responses) or even before a saccade onset (*Colby et al., 1996*; *Kusunoki and Goldberg, 2003*). This so-called forward remapping offers an excellent candidate mechanism of space constancy that has been incorporated into such phenomenon models. In these models, connections between visual neurons with receptive fields spatially separated by the saccade vector can be used to predict post-saccadic visual stimulus location and compensate for retinal image shifts during the saccade (*Neupane et al., 2017*; *Quaia et al., 1998*; *Wang et al., 2016*). However, forward remapping models have been challenged by Zirnsak and colleagues (*Chen et al., 2018*; *Zirnsak et al., 2014*), whose results suggested that the early findings were affected by the low spatial resolution of the receptive field mapping technique (*Duhamel et al., 1992*; *Sommer and Wurtz, 2006*; *Walker et al., 1995*). More detailed FEF receptive field mapping suggested that, before a saccade, cells preferentially respond to stimuli presented near the saccade target rather than at the remapped target location. In other words, cell receptive fields do not shift in parallel to the saccade vector, but instead converge towards the saccade target. Zirnsak and colleagues thus argued that forward remapping models cannot explain space constancy and instead one should focus on saccade target selection as a mechanism mediating this phenomenon (*Zirnsak and Moore, 2014*). More recent work has indicated that the visual system may, in fact, implement both forward and convergent remapping of receptive fields in area V4 (*Neupane et al., 2016b*; *Neupane et al., 2016a*). As this combined approach has been criticized on technical grounds (*Hartmann et al., 2017*), the neurophysiological results regarding the existence of forward remapping remain inconclusive.

On the other hand, a number of behavioral studies have repeatedly and reliably demonstrated forward remapping of spatial attention before saccades (*Jonikaitis et al., 2013*; *Rolfs et al., 2011*; *Szinte et al., 2015*; *Szinte et al., 2016*; *Yao et al., 2016b*). The behavioral studies, however, suffered from the same drawbacks as early neurophysiological studies, low spatial resolution. Our study has eliminated this potential criticism, and our observed deployment of attention in the opposite direction of the saccade and of the cue speaks in favor of the forward remapping effects. Consistent with previous behavioral studies (*Baldauf and Deubel, 2008*; *Deubel and Schneider, 1996*; *Jonikaitis et al., 2017*) and contrary to the convergent remapping effects, our results show that spatial attention is allocated to the saccade target and does not broadly spread around it. Additionally, convergent remapping cannot account for a number of earlier behavioral findings (*Jonikaitis et al., 2013*; *Rolfs et al., 2011*; *Szinte et al., 2015*; *Szinte et al., 2016*), as such spread of attention would have to be asymmetric and not spread towards the several control positions tested in these earlier studies.

Of note, the behavioral consequences of pre-saccadic changes in the spatial tuning of visual cells RF are unclear. Perhaps counterintuitively, recent computational neuroimaging modeling (*Kay et al., 2015*) has shown that increasing the neural spatial sampling at a particular position, similar to over-sampling of the saccade target observed within convergent remapping studies (*Hartmann et al., 2017*; *Tolias et al., 2001*; *Zirnsak et al., 2014*), results in reduction of spatial uncertainty. Convergent remapping does not necessarily yield a large spread of attention around the saccade goal (*Zirnsak and Moore, 2014*). Rather, it may increase visual sensitivity to stimuli only in the immediate vicinity of the saccade target. Convergent remapping could, therefore, reflect the spatially specific attentional selection of the saccade target observed in the present study. Our results indicate that spatial visual attention mechanisms must be accounted for in future work of remapping to advance our understanding of space constancy. Here, we hypothesize that previous reports of convergent

remapping may reflect increased visual sensitivity at the saccade target explained by both spatial and temporal properties of visual attention.

First, to determine a visual neuron RF spatial profile, neurophysiologists used localized visual probes, presented most of the time in a sparse display with high probe-background contrast. It is, therefore, likely that such probes capture spatial attention (*Theeuwes, 1991*). The same holds for visual stimuli used to trigger the saccade, which were both task-relevant and, in most experiments, high-contrast stimuli. As visual RFs can shift towards attended locations even without any saccade involved (*Niebergall et al., 2011*; *Womelsdorf et al., 2006*), one must account for the effect of attention before interpreting any RF change in spatial tuning. In our study, the attention-capturing cue and the saccade target were dissociated from the measure of spatial attention. We measured attention by a discrimination target that did not significantly capture attention on its own, and, therefore, did not interfere with saccade preparation or pre-saccadic distribution of spatial attention (*Deubel and Schneider, 1996*). We also use different conditions to account separately for the effect of the saccade target and of the cue. To understand pre-saccadic RF changes in spatial tuning, one should first consider the spatial deployment of spatial attention.

Second, we argue that the temporal dynamics of attention have to be accounted for. We found that the time at which we presented our cue was critical for remapping of attention to be observed before the saccade. In particular, benefits at the remapped location of the cue were observed only when the cue was shown more than 175 ms before the saccade onset. As even the fastest deployment of attention would take a minimum of 100 ms to occur (*Ling and Carrasco, 2006*; *Müller and Rabbitt, 1989*; *Nakayama and Mackeben, 1989*; *Rolfs and Carrasco, 2012*), our remapping effects are compatible with the time course of attentional selection. Different neurophysiology studies measured visual cell activity followingpresentation of visual objects at different times across saccades (*Kusunoki and Goldberg, 2003*; *Marino and Mazer, 2018*; *Nakamura and Colby, 2002*; *Wang et al., 2016*). In particular, it was reported that a fair proportion of visual neurons recorded within LIP (*Kusunoki and Goldberg, 2003*) and earlier visual areas (*Nakamura and Colby, 2002*) showed forward remapping activity for probes presented as early as 250 ms before the saccade. Interestingly, they observed forward remapping activity preceding the saccade onset only if a visual object was flashed early before the saccade, such that visual objects shown earlier relative to the eye movement resulted in delayed activity during or after the saccade onset (*Kusunoki and Goldberg, 2003*; *Nakamura and Colby, 2002*). Our results are consistent with these findings, but not with those of Wang and colleagues (*Wang et al., 2016*), which report that the presentation of a visual object before the saccade resulted only in post-saccadic forward remapping activities within a set of recorded cells in LIP. Moreover, they found that the recorded cells responded transiently to the intermediate position between the current and the future receptive field position. By contrast, we did not observe any benefit at locations between the cue and the remapped position of the cue. These effects suggest that remapping of attention better reflects the activity of early visual areas than attentional priority maps (LIP, FEF, SC).

It is important to note that neurophysiology studies that failed to observe pre-saccadic forward remapping typically presented probes shortly before (50 ms) the saccade. Our behavioral findings suggest that a short time window between probe presentation and neural recording, which some studies have used, might be insufficient for probes to be remapped to a location parallel to the saccade before the onset of the movement. Further, if remapping is closely related to the time course of attention, it is possible that for attended stimuli shown just prior to the saccade onset, remapping may occur during or after the saccade (*Kusunoki and Goldberg, 2003*; *Nakamura and Colby, 2002*). Indeed, Neupane and colleagues (*Neupane et al., 2016a*; *Neupane et al., 2016b*) observed forward remapping when measuring post-saccadic memory responses to probes shown just before the saccade. Also, Yao and colleagues (*Yao et al., 2016a*) have shown that a post-saccadic memory trace of remapping was enhanced by attentional modulations established before the saccade, corroborating the notion that forward remapping can occur after the saccade. Although we did not measure whether a cue shown shortly before saccade onset was remapped after the saccade in the present study, this was done in two previous studies (*Jonikaitis et al., 2013*; *Yao et al., 2016b*). Both studies found that spatial attention was allocated to the location of a salient stimulus immediately after the saccade, even when the stimulus was no longer present (*Jonikaitis et al., 2013*). This indicates that the visual system anticipates the attended stimulus location after a saccade and recomputes its retinotopic location before the saccade is done. Finally, we also observed high

perceptual performance at fixation, a result in line with two previous studies that investigated foveal remapping effects (*Knapen et al., 2016*; *Rolfs et al., 2011*). Such an attentional effect is surprising, as one would expect that visual selection should prioritize the saccade target, whereas the current fixation should be the least informative and least attended part of the display (especially given that participants already fixated for ~1 s before starting saccade preparation). However, if one considers that fixation-centered receptive fields will process the saccade target after the saccade, forward remapping effects suggest significant attentional benefits at that location, as we observed here and in a previous study (*Rolfs et al., 2011*).

Marino and Mazer (*Marino and Mazer, 2018*) recently showed that attention modulates the state of neurons in V4 before saccade onset, without a substantial shift of neurons' spatial tuning. In contrast to our study, which manipulated transient attention by cueing a location on every trial, they used a sustained attention task to a cue presented before a series of records. Their pre-saccadic effects are consistent with the results reported here and suggest that new models of space constancy should account for the dynamics of spatial attention.

In summary, we used an eccentricity-adjusted discrimination task to measure, for the first time, spatial attention maps before saccades. Using this method, we observed a spatially specific increase in visual sensitivity at the fixation target, the saccade target, the cue and the remapped location of the cue. We found no evidence supporting the convergent remapping interpretation, which suggests that spatial attention spreads around the saccade target in a spatially unspecific way. We found that, before a saccade, attention is deployed towards the saccade target as well as towards a cued location. Further, given that the cue was presented sufficiently early before the saccade, we observed a deployment of attention to its remapped location, that is parallel and opposite to the saccade vector. Although the benefit at that location is smaller compared with that at the cue location, it reflects an ongoing process that facilitates spatial attention allocation after the saccade despite the retinotopic shifts induced by the eye movement.

## Materials and methods

### Participants

Eighteen students (14 participants in the peripheral remapping task, eight participants in the foveal remapping task, four participants did both tasks) of the Ludwig-Maximilians-Universitä München participated in the experiment (ages 22–30, 10 female, one author), for a compensation of 10 Euros per hour of testing. All participants except the author were naive as to the purpose of the study and all had normal or corrected-to-normal vision. The experiments were undertaken with the understanding and written informed consent of all participants and were carried out in accordance with the Declaration of Helsinki. Experiments were designed according to the ethical requirements specified by the Faculty for Psychology and Pedagogics of the LMU München (approval number 13_b_2015) for experiments involving eye tracking. All participants provided written informed consent, including a consent to publish anonymized data.

### Setup

Participants sat in a quiet and dimly illuminated room, with their head positioned on a chin and forehead rest. The experiment was controlled by an Apple iMac Intel Core i5 computer (Cupertino, CA, USA). Manual responses were recorded via a standard keyboard. The dominant eye's gaze position was recorded and available online using an EyeLink 1000 Desktop Mounted (SR Research, Osgoode, Ontario, Canada, RRID:SCR_009602) at a sampling rate of 1 kHz. The experimental software controlling the display, the response collection as well as the eye tracking was implemented in Matlab (MathWorks, Natick, MA, USA, RRID:SCR_001622), using the Psychophysics (*Brainard, 1997*; *Pelli, 1997*) and EyeLink toolboxes (*Cornelissen et al., 2002*). Stimuli were presented at a viewing distance of 60 cm, on a 21-in gamma-linearized Sony GDM-F500R CRT screen (Tokyo, Japan) with a spatial resolution of 1024 × 768 pixels and a vertical refresh rate of 120 Hz.

### Procedure

We completed two different tasks (peripheral remapping and foveal remapping) in a total of four experimental sessions (on different days) of about 100 min each (including breaks). Each task was

always preceded by a discrimination threshold measurement, completed at the beginning of each session. Each session was composed of two blocks of the threshold task followed by four to six blocks of the main task. Participants ran a total of 11–12 blocks of the peripheral remapping task and four blocks of the foveal remapping task. Participants who completed the two tasks always started with the peripheral remapping task.

## Peripheral remapping task

Each trial began with participants fixating a fixation target, a red frame (2.2 dva/side, 10' width, 30 cd/m$^2$) presented on a gray background (60 cd/m$^2$). When the participant's gaze was detected within a 2.0 dva radius virtual circle centered on the fixation target for at least 200 ms, the trial began with a random fixation period of 500–900 ms (uniform distribution, in steps of 50 ms). After this period, the fixation target was replaced by the saccade target (same red frame) presented 12 dva to the right or to the left of the fixation target (*Figure 1A*). Participants were instructed to move their eyes as quickly and as accurately as possible towards the center of the saccade target. From the beginning of the trial, we presented 12 flickering visual streams (40 Hz), composed of 25 ms vertical Gabor patches (frequency: 2.5 cycles per degree; 100% contrast; same random phase on each screen refresh; standard deviation of the Gaussian window: 0.9 dva; mean luminance: 60 cd/m$^2$) alternating with 25 ms pixel noise square masks (2.2 dva side, made of ~0.04 dva-width pixels). The visual streams were arranged in three by four matrix, with a distance of 6 dva between each element (*Figure 1B*). On each trial the matrix of 12 visual streams was presented at one out of 15 different positions relative to the display center (shifted by −6 dva, −3 dva, 0, +3 dva or +6 dva vertically and −3 dva, 0 or +3 dva horizontally). Between 50 and 175 ms after the saccade target onset, one of the 12 vertical Gabor patches was replaced by a discrimination target, a tilted Gabor (clockwise or counterclockwise from the vertical). The time interval of 50–175 ms was determined in a pilot study with two criteria: i) that discrimination target offset occurred in the last 150 ms before the saccade, and ii) taking into account shorter saccade latency on trials when the fixation and saccade targets were not covered by visual streams. Once the discrimination target had appeared, no more vertical Gabor patches were presented and only noise masks alternated with blank frames (*Figure 1B*). Across trials the discrimination target was shown at 32 positions covering 24 dva horizontally and 18 dva vertically (position located at every second intersection of a nine column by seven rows grid, see *Figure 1C–D*). At the end of each trial, participants reported the orientation of the discrimination target using the keyboard (right or left arrow keys), followed by a negative-feedback sound on error trials.

In two-thirds of the trials we captured attention by presenting a task-irrelevant cue, a 50 ms abrupt color onset stimulus presented in between the fixation and saccade targets, 6 dva above or below the screen center. This cue was a green Gaussian patch (mean luminance of 80 cd/m$^2$), with the same Gaussian window of the Gabors and covering one of the visual streams. Across trials this cue was presented either 50 ms or 200 ms before the discrimination target onset. In one-third of the trials the cue was not presented at all. To avoid inter-trial attention-lingering effect at the cue location, we separated cue and no-cue trials, with no-cue trials presented in the first four blocks of the task.

This method allowed us to map the allocation of attention at four positions of interest: the fixation target, the saccade target, the cue, and the remapped position of the cue, as well as 28 control positions. To maximize the number of trials at the different positions of interest, we presented discrimination targets less often (30% less) in the two rows (nine positions) most distant from the cue (e.g. if the cue was presented above the horizontal meridian, the discrimination target was presented less frequently at the two bottom rows below the horizontal meridian). In the trials without a cue, discrimination targets were presented less often (30% less) in the two most peripheral rows (nine top and nine bottom row) maximizing the number trials around the fixation and the saccade target. This procedure was selected mainly to reduce the duration of the experiment. We are confident that reducing the frequency of presenting a discrimination target at these control positions did not affect the deployment of attention. As in previous studies using the same stimulus (*Jonikaitis et al., 2013*; *Rolfs et al., 2011*), we observed at these control positions performance near chance level suggesting that participants are more or less unable to detect the occurrence of the discrimination target. Importantly, we reproduced all the effects presented above, but instead of using all the control positions, we used only those matched in the frequency of discrimination target across trials.

Participants completed between 2914 and 3323 trials of the peripheral remapping task. We checked correct fixation maintenance and correct saccade execution online and repeated incorrect trials at the end of each block. We also repeated trials during which a saccade started within the first 25 ms or ended after more than 350 ms following the saccade target onset (participants repeated between 159 and 443 trials). On average, we analyzed 25.38 ± 0.76 trials and 16.13 ± 0.42 trials per participant at the frequently and less frequently tested positions, respectively, in each of the three main conditions.

## Peripheral remapping threshold task

On each session, before the peripheral remapping task, participants completed a threshold task. This allowed us to avoid possible effects of task learning across different sessions as well as to adjust the discrimination target tilt for different eccentricities from fixation. This latter point is particularly important as it reduced the impact of visual acuity (*Paradiso and Carney, 1988*) on the measure of spatial attention. The threshold task was identical to the main task with the exception that participants kept fixation and the saccade target was not shown. After a random initial period of 500–900 ms (uniform distribution, in steps of 50 ms), a cue was briefly (50 ms) shown followed by a discrimination target 200 ms later at the cued location. Across trials the cue was shown at each of the 32 locations used in the main experiment. The positions of discrimination target and cue were subdivided into five equiprobable groups of eccentricity from the fixation target (eccentricity 1: the fixation target; eccentricity 2: from ~4.2 to 6 dva, eccentricity 3: from ~8.5 to ~9.5 dva; eccentricity 4: from 12 to ~13.4 dva; and eccentricity 5: from ~15.3 to ~16.2 dva). We used a procedure of constant stimuli and a randomly selected orientation of the discrimination target from a linearly spaced interval for each eccentricity (each interval divided into five steps; between ±1 and ±9 dva for the first eccentricity, between ±1 and ±13 dva for eccentricity two and between ±1 and ±17 dva for eccentricities three to five).

Participants were instructed that the cue indicated the position of the discrimination target and were told to report at the end of each trial its orientation (clockwise or counterclockwise). They completed 400 trials across two blocks of the threshold task. For each participant and experimental session individually, we determined for the five eccentricities from the fixation target, five threshold values corresponding to the discrimination target tilts leading to a correct discrimination in 85% of the trials. To do so, we fitted five cumulative Gaussian functions to performance gathered in the threshold blocks. These tilts were used in the main task at their respective eccentricity from the fixation target. In the main task, only trials in which the discrimination target was presented within 150 ms before the saccade were analyzed. As during this period participants are preparing the saccade, any change of orientation sensitivity over space is attributed to the saccadic planning and/or to localized deployment of attention.

## Foveal remapping and threshold tasks

The foveal remapping task was identical to the peripheral remapping task, with the exception that we did not present the cue. Moreover, the foveal remapping threshold task differed from the peripheral remapping threshold task, as participants made a saccade during the threshold task instead of keeping fixation (*Rolfs et al., 2011*). In the foveal remapping threshold task the saccade target could be presented at any of the 32 locations tested. The discrimination target appeared 200 ms after the appearance of the saccade target and participants were instructed that the discrimination target could appear at either fixation or saccade target. We again used a procedure of constant stimuli, and chose discrimination target orientation randomly for intervals defined for the different eccentricities (intervals divided into five steps for each eccentricity; intervals between ±1 and ±25 dva for the eccentricities one and two and between ±1 and ±21 dva for eccentricities three to five).

Participants completed between 973 and 1137 trials of the foveal remapping main task. We checked correct fixation maintenance and correct saccade execution online and repeated incorrect trials at the end of each block. We also repeated trials during which a saccade started within the first 25 ms or ended after more than 350 ms following saccade target onset (participants repeated between 13 and 177 trials). On average, we analyzed 32.12 ± 0.82 trials and 15.73 ± 0.64 trials per participant at the frequently and less frequently tested positions, respectively. Participants completed 500 trials across two blocks of the threshold task. For each participant and experimental

session individually, we determined for the five eccentricities from the fixation target, five threshold values corresponding to the angles leading to correct orientation discrimination in 85% of the trials following the same procedure as in the peripheral threshold task.

## Data pre-processing

Recorded eye position data were processed offline (independent of online tracking during the experiment). Saccades were detected based on their velocity distribution (*Engbert and Mergenthaler, 2006*) using a moving average over 20 subsequent eye position samples. Saccade onset was detected when the velocity exceeded the median of the moving average by 3 SDs for at least 20 ms. We included trials if a correct fixation was maintained within a 2.0 dva radius centered on the fixation target, if a correct saccade started at the fixation target and landed within a 2.0 dva radius centered on the saccade target, and if no blink occurred during the trial. Finally, only trials in which the discrimination target disappeared in the last 150 ms preceding saccades were used in the analysis. In total, we included 36,236 trials (90.41% of the online accepted trials, 82.20% of all trials) in the peripheral remapping main task and 7306 trials (95.13% of the online accepted trials, 86.85% of all trials) of the foveal remapping main task.

## Behavioral data analysis

Data were analyzed separately for the three conditions of the peripheral remapping task and the only condition of the foveal remapping task. For the trials in which a cue was presented, the cue onset preceded the discrimination target onset by either 50 or 200 ms. Therefore, one condition included the trials with a SOA of 50 ms during which the cue disappeared in the last 175 ms before the saccade, and a second condition included the trials with a SOA of 200 ms during which the cue disappeared more than 175 ms before the saccade. A third condition of the peripheral remapping task and all the trials of the foveal remapping task included trials in which no cue was shown.

To map the allocation of attention, we first mirrored discrimination target positions of leftward saccade trials to match those of the rightward saccade trials. Moreover, in trials with a cue, we mirrored positions of the bottom cue trials (trials in which the cue was shown 6 dva below the screen center) to match those of the top cue trials. Then, for each participant and each condition, we determined the sensitivity in discriminating the orientation of the discrimination target (d'): $d'=z(hit\ rate) - z(false\ alarm\ rate)$. To do so, we defined a clockwise response to a clockwise discrimination target (arbitrarily) as a hit and a clockwise response to a counterclockwise discrimination target as a false alarm. Corrected performance of 99% and 1% were substituted if the observed proportion correct was equal to 100% or 0%, respectively. Performance values below the chance level (50% or d' = 0) were transformed to negative d' values. We next normalized for each participant individually, the sensitivity obtained at each position by the range obtained across all tested positions following this formula $d'_n = (d'_n - min) / (max - min)$, with $d'_n$ the sensitivity at a given $n$ position, $min$ and $max$, respectively, the minimum and maximum sensitivity obtained across the 32 tested positions in the specific condition. These normalized values were then averaged across participants and used to plot sensitivity maps and to perform statistical comparisons. Subtraction maps (*Figure 3*) were obtained by first subtracting the normalized sensitivity difference between trials in which the cue was presented 50 ms or 200 ms before the discrimination target onset from trials in which no cue was shown. For this comparison we used normalized sensitivity of raw sensitivity as the conditions differ both in the number of trials per discrimination target and in the number of expected salient locations across conditions. These difference values were then normalized across the position according to the same formula as above and averaged across participants to highlight differences between the two obtained subtraction maps.

We obtained sensitivity maps (*Figures 2B, E, H*, *3A, B* and *4A*), by interpolating (triangulation-based natural neighbor interpolation) the missing values located at every second intersection of the nine columns by seven rows grid of discrimination targets using the 32 tested positions. We then rescaled the grid (Lanczos resampling method) to obtain a finer spatial grain. We drew position sensitivity maps across participants as colored maps coding the mean sensitivity across participants following a linear color scale going from 0.25 to 0.75 normalized sensitivity. To determine the threshold maps (*Figure 1C–D*), we first interpolated (linear interpolation) the mean threshold angle obtained for each participant individually over the five different distances between the fixation

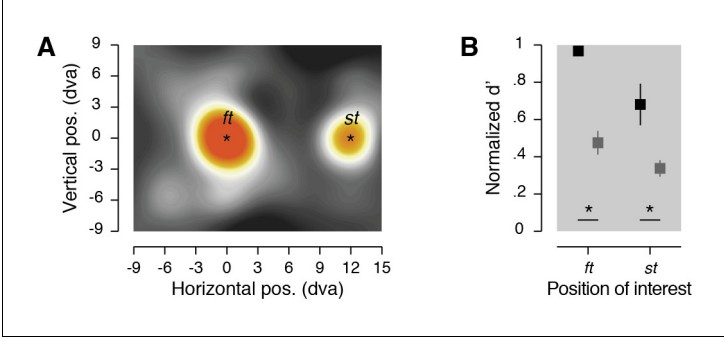

**Figure 4.** Foveal remapping task results. (**A**) Normalized sensitivity maps. Averaged normalized sensitivity (d'). (**B**) Averaged normalized sensitivity (d')obtained at two positions of interest (see center in black) and at their corresponding surround positions (see surround in dark gray). Conventions and color scale are as in *Figure 2*.
DOI: https://doi.org/10.7554/eLife.37598.005

target and the discrimination target. Map of threshold *dt* angles across participants was then obtained by drawing disks centered on the fixation target with a radius corresponding to the eccentricity at which the discrimination target was played and coded the mean threshold angle obtained across participants following a linear color scale going from 0° to 20° of discrimination target tilt.

For statistical comparisons we drew (with replacement) 10000 bootstrap samples from the original pair of compared values. We then calculated the difference of these bootstrapped samples and derived two-tailed *p* values from the distribution of these differences. Statistical comparisons of the eccentricity effect of the discrimination target on different saccade metrics (saccade latency and accuracy) were tested using repeated measures ANOVA. Discrimination target positions were grouped depending on their eccentricities from the fixation target as defined in the threshold tasks.

## Acknowledgements

This research was supported by a Deutsche Forschungsgemeinschaft temporary position for principal investigator grant to MS (SZ343/1) and DR (RA2191/1-1), a Marie Sklodowska-Curie Action Individual Fellowship to MS (704537). We are grateful to the members of the Deubel laboratory in Munich for helpful comments and discussions and to Elodie Parison, Alice and Clémence Szinte for their invaluable support.

## Additional information

### Funding

| Funder | Grant reference number | Author |
|---|---|---|
| Deutsche Forschungsgemeinschaft | SZ343/1 | Martin Szinte |
| Deutsche Forschungsgemeinschaft | RA2191/1-1 | Dragan Rangelov |
| H2020 Marie Skłodowska-Curie Actions | 704537 | Martin Szinte |

The funders had no role in study design, data collection and interpretation, or the decision to submit the work for publication.

### Author contributions

Martin Szinte, Conceptualization, Resources, Data curation, Software, Formal analysis, Funding acquisition, Investigation, Visualization, Methodology, Writing—original draft, Project administration, Writing—review and editing; Donatas Jonikaitis, Conceptualization, Investigation, Methodology, Writing—original draft, Writing—review and editing; Dragan Rangelov, Conceptualization,

Resources, Investigation, Methodology, Writing—review and editing; Heiner Deubel, Conceptualization, Supervision, Methodology, Project administration, Writing—review and editing

## Author ORCIDs
Martin Szinte (iD) http://orcid.org/0000-0003-2040-4005
Donatas Jonikaitis (iD) http://orcid.org/0000-0001-9851-0903

## Ethics
Human subjects: Experiments were designed according to the ethical requirements specified by the Faculty for Psychology and Pedagogics of the Ludwig-Maximilians-Universität München (approval number 13_b_2015) for experiments involving eye tracking. All participants provided written informed consent, including a consent to publish anonymized data.

## Decision letter and Author response
Decision letter https://doi.org/10.7554/eLife.37598.010
Author response https://doi.org/10.7554/eLife.37598.011

## Additional files

### Supplementary files
• Transparent reporting form
DOI: https://doi.org/10.7554/eLife.37598.006

### Data availability
All files are available from the OSF database: URL: https://osf.io/3tru6.

The following dataset was generated:

| Author(s) | Year | Dataset title | Dataset URL | Database and Identifier |
| --- | --- | --- | --- | --- |
| Szinte M, Jonikaitis D, Rangelov D | 2018 | Dataset of the Peripheral Remapping and Foveal Remapping of attention tasks | https://osf.io/3tru6 | Open Science Framework, osf.io/3tru6 |

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
