## [Decision Letter]

Thank you for submitting your article "Pre-saccadic remapping relies on dynamics of spatial attention" for consideration by *eLife*. Your article has been reviewed by three peer reviewers, and the evaluation has been overseen by Sabine Kastner as the Reviewing and Senior Editor. The reviewers have opted to remain anonymous.

The reviewers have discussed the reviews with one another and the Reviewing Editor has drafted this decision to help you prepare a revised submission.

Summary

The study conducted by Szinte et al. attempts to resolve the debate whether the spatial updating mechanism linked to saccade planning and execution that was discovered by neurophysiological studies has a psychophysical counterpart. The experiment was designed on the premise that spatial updating operates on locations in the visual field where attention is deployed, such as the target of an eye movement or an attention-soliciting cue. Using a high-resolution mapping procedure coupled with a visual discrimination task, the authors tried to determine if psychophysical thresholds are reduced (i.e. visual discrimination improves) in three key regions where attentional resources could be allocated: 1. the position of the saccade target, 2. that of an attention cue and, critically, 3. the "remapped" location of that cue, i.e. the retinotopic location it will occupy as a consequence of the saccade. Consistent with previous work, discrimination performances were found to be improved in close vicinity of the saccade target and of the attention cue. Furthermore, the results provide support in favor of the forward remapping hypothesis, by showing a small but significant enhancement of discrimination performance at a location distant from the attention cue but which corresponds precisely to the retinotopic receptive field of the visual neurons that will encode this cue at the end of the saccade. Critically, the effect is observed only if the cue is presented more than 200ms before the saccade, indicating that attentional instruction has to be fully processed for the remapping to take place.

All reviewers agreed that this is a carefully and timely study and makes an important contribution to the ongoing debate on spatial remapping.

Essential revisions:

1) Experimental design concerns. In particular, it was unclear how the eccentricity-based orientation thresholding was applied. While the benefits of the design were appreciated, it is critical that these thresholds be properly selected, otherwise it might affect the pattern of results, because the analysis is directly comparing sensitivity across locations that had different difficulty levels. In order to address this concern, the sensitivity at each location should be compared to the baseline no-cue condition at that location (i.e., subtract Figure 2B from 2E and 2F). This would alleviate concerns about directly comparing across locations that may not have been properly equated. It would also resolve a related concern, which is that in the Materials and methods it is revealed that not all target locations were probed with the same frequency. This suggests that it may be misleading to test sensitivity at a given location relative to the average of all tested locations; rather, it should be compared to the average of all locations probed with the same frequency – or better yet, to that exact location under the no-cue baseline condition.

This essential analysis would also alleviate a related point, that is, the timing of forward remapping does not match with neurophysiology. Firstly, in paragraph three of the Introduction, the authors mention that "contrary to neurophysiology", they manipulated the timing of stimulus onset relative to the saccade. There are, however, neurophysiological studies that explicitly looked at remapping vs timing of probes (Kusunoki and Goldberg, 2003; Nakamura and Colby, 2002; Wang, et al., 2016). The authors should correct this sentence. Secondly, all of these neurophysiology studies show that the strength of forward remapping increases with decreasing probe-to-saccade onset temporal distance. This apparent mismatch between the authors' observation and neurophysiology could be because they didn't account for changing discrimination threshold across pre-saccadic time period either due to changing visual acuity (Campbell and Wurtz, 1978) or changing strength of spatial attention (Figure 1 C,D). Thirdly, it could simply be that forward remapping could have been observed if probes were flashed after the saccade (similar to what was done by Jonikitis et al., 2013; we acknowledge that the authors do mention this in discussion). These different possibilities should be discussed in the context of the results from the control analysis.

2) Some of the conclusions should be toned down, and the framing should be clarified:

i) The authors appear to over-reach in the implications of this study for the neural receptive field remapping debate. The neural debate is whether receptive fields spatially shift in a forward or convergent manner. The current study is a behavioral study of attention. The authors find some very interesting results, but it is unclear whether any definitive conclusions can be drawn about visual receptive field remapping from a behavioral study of attention. E.g., the conclusion at the end of the Abstract is that "pre-saccadic remapping is an attentional process […]". It is not clear how these results show that remapping in general is an attentional process, or that the conclusions about the nature of attentional remapping necessarily generalize to remapping in general. It may be preferable to contextualize these results in the context of "remapping of attention" only, and to include some discussion of studies that have debated whether the process of remapping involves receptive fields shifting spatially vs whether remapping is more of an attentional process (e.g. Cavanagh et al., 2010; Marino and Mazer, 2018).

ii) The authors rightly use multiple probe locations to measure remapping (the novelty of this paper and a much needed experiment in the field). But they might be looking at high spatial resolution at the cost of low temporal resolution (a caveat also in Zirnsak et al., 2014). In this study, the caveat is the choice of distractor target (DT) orientation despite a changing threshold for it across time. They are comparing conditions for probes presented long before and shortly before the saccade using the same DT orientation. This may be the reason that no remapping was observed when probes were flashed near the saccade onset. In other words, why isn't it the case that forward remapping wasn't observed for late probes because the DT discrimination threshold was higher then? Indeed visual acuity is known to change near the saccade onset (Campbell and Wurtz, 1978) and the authors also show that the threshold is modulated by saccade (Figure 1C vs D and foveal remapping results in Figure 3). These caveats need to be discussed.

iii) While the study seems to demonstrate convincing evidence in favor of forward remapping, the reviewers and editors were specifically uncomfortable with the strong tone of this paper on the 'lack of convergent remapping'. We think that this particular conclusion will need revision.

[Editors' note: further revisions were requested prior to acceptance, as described below.]

Thank you for resubmitting your work entitled "Pre-saccadic remapping relies on dynamics of spatial attention" for further consideration at *eLife*. Your revised article has been favorably evaluated by Andrew King (Senior Editor), a Reviewing Editor, and two reviewers.

The manuscript has been improved but there are some remaining issues that need to be addressed before a final decision can be made, as outlined by reviewer 1 below:

*Reviewer #1:*

While I still think the results make an interesting and important contribution to the field (and I am reassured by the no-cue-baseline subtraction analysis), I found the revision unsatisfactory.

1) The strong anti-convergent remapping tone of the paper has not really been toned down. There are places where it is improved, but other places where it seems like the authors have doubled-down on how it "indisputably rejects" that hypothesis.

2) The sentence in the Abstract that was flagged as being problematic for claiming that "pre-saccadic remapping is an attentional process" has not been changed. Moreover, while the discussion is a bit more careful about linking the behavioral and neural results, the tone is less one of framing the current results in terms of remapping of attention (and acknowledging potential limitations of the behavioral approach), and more one of repeatedly criticizing the neural studies for not taking attentional effects into account. This is a fair criticism that absolutely should be mentioned, but it's not enough to just do that.

3) The new subtraction analysis that was presented as an "essential revision" is hidden in the supplement (why? If anything it makes their results even more salient), and there's not enough detail provided. It's not clear what exactly the comparisons are for the statistics given in paragraph five of the Results section: Are these simply t-tests between the normalized d' at the remapped location for no-cue vs early-cue and no-cue vs late cue? If so, what does it mean that "these results were evident even after subtracting the normalized sensitivity"? Moreover, what about the other stats presented for the main analyses? The supplemental figure only shows normalized sensitivity maps, not the plots of d' at the positions of interest and their surround positions. Finally, the Materials and methods themselves are unclear. In subsection “Behavioral data analysis” it states that subtraction maps were obtained by "subtracting the normalized sensitivity difference". First, I'm assuming they meant subtracting the normalized sensitivity "values"? (I.e., it wasn't actually a subtraction of difference scores?) Second, what is the rationale for the 2-step normalization: subtracting the normalized sensitivities and then doing a second normalization on those difference values? Wouldn't it be better to subtract the raw sensitivity scores (and then only apply the normalization on the difference maps)?

*Reviewer #3:*

The authors have addressed all the concerns. I am satisfied with the paper and it qualifies for publication. I do have some reservations with their interpretation of Kusunoki and Goldberg, 2003, Nakamura and Colby, 2002, and Wang et al., 2016, but my reservations are beyond the scope of this paper.

---

## [Author Response]

Essential revisions:1) Experimental design concerns. In particular, it was unclear how the eccentricity-based orientation thresholding was applied. While the benefits of the design were appreciated, it is critical that these thresholds be properly selected, otherwise it might affect the pattern of results, because the analysis is directly comparing sensitivity across locations that had different difficulty levels. In order to address this concern, the sensitivity at each location should be compared to the baseline no-cue condition at that location (i.e., subtract Figure 2B from 2E and 2F). This would alleviate concerns about directly comparing across locations that may not have been properly equated. It would also resolve a related concern, which is that in the Materials and methods it is revealed that not all target locations were probed with the same frequency. This suggests that it may be misleading to test sensitivity at a given location relative to the average of all tested locations; rather, it should be compared to the average of all locations probed with the same frequency – or better yet, to that exact location under the no-cue baseline condition.

We extended and clarified our description of the threshold procedure in the Materials and method and Results sections. We believe our threshold procedure was necessary to evaluate maps of sensitivity without eccentricity-based effects. Indeed, the absence of an eccentricity gradient in the observed sensitivity maps suggests that this procedure was successful (see Figures 2B, 2E, 2H and Figure 3A). However, we agree with the reviewers that these results cannot guarantee that the threshold procedure had no other effect on our results. As suggested by them, we prepared maps of sensitivity difference between the conditions (see Figure 2—figure supplement 1 and paragraph five of the Results and subsection “Behavioral data analysis”). The obtained maps correspond to what one would expect from a visual subtraction of the Figure 2E and 2H from the Figure 2B, with a presaccadic remapping of attention effect observed only when subtracting the early cue (Figure 2H) condition to the no cue (Figure 2B) condition. Next, as suggested by the comments of the reviewers, we directly compared the position of interest to the “baseline” no cue condition and found results directly compatible with our former analyses and conclusions. In particular, we observed a benefit at the remapping position of the cue only when the no cue condition was compared to the early cue but not to the late cue condition. We now report these results in the revised manuscript (paragraph five of the Results).

From the second part of the comment, we understood that reviewers were concerned about the fact that we compared individual positions of interest to the average of all positions, while some positions were less frequently tested. As pointed out by the reviewers, we did not use the same frequency of testing at all positions, principally to spare our participants the extra effort in an already very long experiment. We reduced the frequency of testing only at positions in which we had good reason to believe participants will not discriminate the targets well. Indeed, we showed in our earlier work (e.g. Rolfs et al., 2011; Jonikaitis, et al. 2013; Wollenberg, et al. 2018) that at similar control positions, sensitivity is close to chance level and overall that the presence of a discrimination target has no observed effect on saccade preparation. In the present study, we replicate these effects and now discuss on these aspects in the revised manuscript (see subsection “Peripheral remapping task”). Nevertheless, following the reviewers’ suggestion, we re-analyzed all the effects reported in the submitted manuscript by comparing each position of interest to all locations assessed with the same frequency. We did not observe any change in the statistics both for the peripheral and the foveal remapping tasks. This reanalysis of our data suggests that our results can’t be explained by a difference in discrimination target frequency across the tested positions. We believe that the discrimination target would have to capture attention for frequency of testing different locations to have an effect on attentional deployment. Our analyses of saccadic latency and saccade accuracy and performance strongly suggest that the location of discrimination target did not, in fact, capture attention. With this in mind, and in the light of the novel, control analyses, we found that the potential impact of testing frequency is minimal. For these reasons, we chose to keep the original analyses in the manuscript and we now mention the novel, control analyses (see subsection “Peripheral remapping task”).

This essential analysis would also alleviate a related point, that is, the timing of forward remapping does not match with neurophysiology. Firstly, in paragraph three of the Introduction, the authors mention that "contrary to neurophysiology", they manipulated the timing of stimulus onset relative to the saccade. There are, however, neurophysiological studies that explicitly looked at remapping vs timing of probes (Kusunoki and Goldberg, 2003; Nakamura and Colby, 2002; Wang, Goldberg et al., 2016). The authors should correct this sentence. Secondly, all of these neurophysiology studies show that the strength of forward remapping increases with decreasing probe-to-saccade onset temporal distance. This apparent mismatch between the authors' observation and neurophysiology could be because they didn't account for changing discrimination threshold across pre-saccadic time period either due to changing visual acuity (Campbell and Wurtz, 1978) or changing strength of spatial attention (Figure 1 C,D). Thirdly, it could simply be that forward remapping could have been observed if probes were flashed after the saccade (similar to what was done by Jonikitis et al., 2013; we acknowledge that the authors do mention this in discussion). These different possibilities should be discussed in the context of the results from the control analysis.

We modified the manuscript and added a section discussing the mentioned papers and their outcomes in regard of our results (see Discussion section). However, we believe that there may be a source of conceptual confusion at play when drawing analogies between psychophysical and neurophysiological studies.

First, we used the term "cue" to refer to a peripheral "probe" that may subsequently be remapped. The cue stimulus is analogous to the electrophysiology probes (flash lights, squares, bars, etc.) shown either inside or outside the recorded pre-saccadic cell’s receptive field. To evaluate the deployment of attention, we measured response accuracy to a "discrimination target" presented before the saccade. In electrophysiology, using discrimination targets is not mandatory as neurophysiologists can directly record cell activity. We observed when the cue (probe) was shown long before the saccade (early cue), a pre-saccadic deployment of attention at the remapped position of the cue. This result is in agreement to what Nakamura and Colby, 2002, and Kusunoki and Goldberg, 2003, observed when they presented probes long before the saccade. If the probe was presented at the future receptive field position, they reported an increase in cells’ activity starting before the execution of the saccade. This effect was observed both for neurons recorded within the brain features maps (V3A, V3, V2, V1) and the brain priority maps (LIP).

When the probe was presented immediately prior to the execution of the saccade (corresponding to our *late cue* condition) the electrophysiological studies found delayed neural responses, starting either during or after the saccade. As the neurons, in the late probe condition, were unresponsive before the saccade, one should not expect benefits at the remapped position before the saccade. Rather, the remapping benefits should be observed after the saccade. We proposed this explanation already in our submitted manuscript. In the revised manuscript we discuss these issues in more details explicitly relating our findings to the electrophysiological studies (see Discussion section).

The results we observed are, in our opinion, in close agreement with the findings of neurophysiology. They also match with a recent study of Marino and Mazer, 2018, who found a transfer of attentional modulation (hand-off) preceding the saccade onset. It is important to note that in their study the allocation of attention is manipulated by cueing a position before a set of recoding trials. The cue of this study could correspond to our early cue condition and their results match with our observed effects. Finally, our results mismatch with the results of Wang and colleagues (2016). These authors report that LIP remapping activity follows the saccade onset for probes presented both long before and just before the saccade. They, moreover, found that the recorded cell’s respond transiently to the intermediate position between the current and the future receptive field position. We do not observe any benefit in between the cue and the remapped position of the cue. These effects suggest that the remapping of attention might better reflect features than priority maps neurophysiology. We discuss these different studies in the revised manuscript.

2) Some of the conclusions should be toned down, and the framing should be clarified:i) The authors appear to over-reach in the implications of this study for the neural receptive field remapping debate. The neural debate is whether receptive fields spatially shift in a forward or convergent manner. The current study is a behavioral study of attention. The authors find some very interesting results, but it is unclear whether any definitive conclusions can be drawn about visual receptive field remapping from a behavioral study of attention. E.g., the conclusion at the end of the Abstract is that "pre-saccadic remapping is an attentional process […]". It is not clear how these results show that remapping in general is an attentional process, or that the conclusions about the nature of attentional remapping necessarily generalize to remapping in general. It may be preferable to contextualize these results in the context of "remapping of attention" only, and to include some discussion of studies that have debated whether the process of remapping involves receptive fields shifting spatially vs whether remapping is more of an attentional process (e.g. Cavanagh et al., 2010; Marino and Mazer, 2018).

In the revised manuscript, we frame our results in the context of the “remapping of attention”, toning down different sentences and mainly discussing the receptive field remapping in the Discussion section. In particular we include a discussion of the study of Marino and Mazer (2018, published after the initial submission) who proposed that remapping of attention may operate without a shift of visual receptive fields.

ii) The authors rightly use multiple probe locations to measure remapping (the novelty of this paper and a much needed experiment in the field). But they might be looking at high spatial resolution at the cost of low temporal resolution (a caveat also in Zirnsak et al., 2014). In this study, the caveat is the choice of distractor target (DT) orientation despite a changing threshold for it across time. They are comparing conditions for probes presented long before and shortly before the saccade using the same DT orientation. This may be the reason that no remapping was observed when probes were flashed near the saccade onset. In other words, why isn't it the case that forward remapping wasn't observed for late probes because the DT discrimination threshold was higher then? Indeed visual acuity is known to change near the saccade onset (Campbell and Wurtz, 1978) and the authors also show that the threshold is modulated by saccade (Figure 1C vs D and foveal remapping results in Figure 3). These caveats need to be discussed.

As the reviewers, we also believe that a measure of sensitivity with high spatial resolution was needed for the field and acknowledge that it comes at the cost of a temporal resolution. We opted for a high-spatial resolution to obtain sensitivity maps for discrimination target presented in the last 150 ms before saccade. This allowed us to compare behavior effects with the controversial records of Zirnsak et al., 2016, in which a similar analysis was used. We also opted for a high spatial resolution, as we already reported in previous studies measures with a fine temporal resolution extending both before (Rolfs et al., 2011, Szinte et al., 2015, Szinte et al., 2016) and after the saccade onset (Jonikaitis et al., 2013). Indeed, combining in the same study a fine spatial and temporal resolution of sensitivity would have involved the collection of at least three to five times the number of trials we had, giving about 15h to 25h of testing per participants.

Next, Campbell and Wurtz, 1978, measured acuity threshold across saccades using a Snellen test chart presented in between a fixation and a saccade target. They reported an elevation of the minimal visible angle at these positions in the last 150 ms before the saccade. We observed the same elevation of the threshold at these positions (see Figure 1C and 1D) and mention this study in the revised manuscript. This finding motivated our foveal remapping task in which we focus on the remapping in between the saccade and the fixation target. Importantly, as we used in the peripheral remapping task the same threshold for each of our conditions, and used for both, no cue, early cue and late cue conditions measures obtained with discrimination targets presented at the same time relative to the saccade onset (last 150 ms), any spatially non-specific change of threshold should have impacted equally our pattern of results. In other words, if as suggested by the reviewers and by Campbell and Wurtz, 1978, the acuity threshold would overall increase before the saccade, this effect can’t explain the difference obtained between the conditions, and especially can’t explain the observed lowering of orientation threshold at the remapped position of the cue. We now specify these aspects in the revised manuscript.

iii) While the study seems to demonstrate convincing evidence in favor of forward remapping, the reviewers and editors were specifically uncomfortable with the strong tone of this paper on the 'lack of convergent remapping'. We think that this particular conclusion will need revision.

We toned down our interpretations on the “lack of convergent remapping”, that we now suggest as one alternative explanation of the observed behavioral findings.

[Editors' note: further revisions were requested prior to acceptance, as described below.]

Reviewer #1:While I still think the results make an interesting and important contribution to the field (and I am reassured by the no-cue-baseline subtraction analysis), I found the revision unsatisfactory.1) The strong anti-convergent remapping tone of the paper has not really been toned down. There are places where it is improved, but other places where it seems like the authors have doubled-down on how it "indisputably rejects" that hypothesis.

We changed our manuscript to temper our conclusion.

2) The sentence in the Abstract that was flagged as being problematic for claiming that "pre-saccadic remapping is an attentional process" has not been changed. Moreover, while the discussion is a bit more careful about linking the behavioral and neural results, the tone is less one of framing the current results in terms of remapping of attention (and acknowledging potential limitations of the behavioral approach), and more one of repeatedly criticizing the neural studies for not taking attentional effects into account. This is a fair criticism that absolutely should be mentioned, but it's not enough to just do that.

We modified the Abstract and edited our manuscript to acknowledge potential limitations of the behavioral approach (see Discussion paragraph two). We make it very clear that our behavioral approach does not impact interpretation of earlier neurophysiology work. However, as earlier neurophysiology work explicitly proposed effects of convergent remapping on attention, we discuss in detail attentional effects and compare our results with other studies.

3) The new subtraction analysis that was presented as an "essential revision" is hidden in the supplement (why? If anything it makes their results even more salient), and there's not enough detail provided. It's not clear what exactly the comparisons are for the statistics given in paragraph five of the Results section: Are these simply t-tests between the normalized d' at the remapped location for no-cue vs early-cue and no-cue vs late cue? If so, what does it mean that "these results were evident even after subtracting the normalized sensitivity"? Moreover, what about the other stats presented for the main analyses? The supplemental figure only shows normalized sensitivity maps, not the plots of d' at the positions of interest and their surround positions. Finally, the Materials and methods themselves are unclear. In subsection “Behavioral data analysis” it states that subtraction maps were obtained by "subtracting the normalized sensitivity difference". First, I'm assuming they meant subtracting the normalized sensitivity "values"? (I.e., it wasn't actually a subtraction of difference scores?) Second, what is the rationale for the 2-step normalization: subtracting the normalized sensitivities and then doing a second normalization on those difference values? Wouldn't it be better to subtract the raw sensitivity scores (and then only apply the normalization on the difference maps)?

We acknowledge that the revised manuscript was somewhat short on details of this condition comparison analyses and therefore understand Reviewer 1’s concerns. We now provide more details (see Results paragraph six and subsection “Behavioral data analysis”) and include the supplementary figure as a main figure (Figure S1 becomes Figure 3).

First, we would like to clarify the statistical comparisons used. These comparisons, as well as all comparisons in the manuscript, with the exceptions of ANOVA, are obtained by drawing with replacement 10,000 bootstrap samples from the original pair of compared values and later calculate the difference of these bootstrapped samples to derive two-tailed *p* values from the distribution of these differences. This procedure is described in the Materials and method section.

We deleted the sentence “these results were evident even after subtracting the normalized sensitivity”, which wasn’t clear.

The main comparison that, we believe, speaks to the “*essential revision*” of Reviewer 1 is described in paragraph six of the Results of the revised manuscript. Briefly, we compared normalized sensitivity obtained at the remapped position of the cue when the cue was presented, with normalized sensitivity obtained at the same position when no cue was shown. We did that analysis and the subtraction figures using normalized values as we understood from the previous review that it was the reviewers’ request (“subtract Figure 2B to Figure 2E and Figure 2F”, which are figures of normalized effects). Importantly, this analysis gives the same effects when instead of comparing normalized d’, one uses comparison of the raw d’ values. In particular we found that sensitivity obtained at the remapped position of the cue in the condition in which no cue was shown (0.01 ± 0.07) significantly differs to the raw sensitivity observed at the same position when the cue appeared substantially before the discrimination target and the saccade onset (0.31 ± 0.09, *p* < 0.0066) but not if it appeared later (0.12 ± 0.14, *p* = 0.4828).

Please, note that we do not present in Figure 3 the comparison of center and surround positions, and no longer present the statistics of single position comparison to all other positions as such comparisons go beyond the scope of the analysis that this figure visualizes.

To be consistent with the use of normalized sensitivity in Figure 2 and the main analysis, we would like to keep Figure 3 with the subtraction of the normalized effects. These normalization steps, and the second step, the normalization of the difference, allows to use the same color scale for all figures (Figure 3A vs. Figure 3B) and to visually compare these subtractions maps to the main effects (Figure 3A-3B vs Figure 2B2E-2H).